# Biocidal Effectiveness of Selected Disinfectants Solutions Based on Water and Ozonated Water against *Listeria monocytogenes* Strains

**DOI:** 10.3390/microorganisms7050127

**Published:** 2019-05-10

**Authors:** Krzysztof Skowron, Ewa Wałecka-Zacharska, Katarzyna Grudlewska, Agata Białucha, Natalia Wiktorczyk, Agata Bartkowska, Maria Kowalska, Stefan Kruszewski, Eugenia Gospodarek-Komkowska

**Affiliations:** 1Department of Microbiology, Nicolaus Copernicus University in Toruń, L. Rydygier Collegium Medicum in Bydgoszcz, 9 M. Skłodowska-Curie St., 85-094 Bydgoszcz, Poland; katinkag@gazeta.pl (K.G.); agatabialucha@wp.pl (A.B.); natalia12127@gmail.com (N.W.); agata.bartkowska@o2.pl (A.B.); gospodareke@cm.umk.pl (E.G.-K.); 2Department of Food Hygiene and Consumer Health, Wrocław University of Environmental and Life Sciences, 31 C.K. Norwida St., 50-375 Wrocław, Poland; ewa.walecka@upwr.edu.pl; 3Department of Food Analytics and Environmental Protection, Faculty of Chemical Technology and Engineering, UTP University of Science and Technology, Seminaryjna 3, 85-326 Bydgoszcz, Poland; maria.kowalska@utp.edu.pl; 4Biophysics Department, Faculty of Pharmacy, Collegium Medicum of Nicolaus Copernicus University, Jagiellońska 13-15 St., 85–067 Bydgoszcz, Poland; skrusz@cm.umk.pl

**Keywords:** *Listeria monocytogenes*, ozon, ozonated water, non-ozonated water, disinfectants, biocidal effectiveness

## Abstract

The aim of this study was to compare the biocidal effectiveness of disinfectants solutions prepared with ozonated and non-ozonated water against *Listeria monocytogenes*. Six *L. monocytogenes* strains were the research material (four isolates from food: meat (LMO-M), dairy products (LMO-N), vegetables (LMO-W), and fish (LMO-R); one clinical strain (LMO-K) and reference strain ATCC 19111). The evaluation of the biocidal effectiveness of disinfectant solutions (QAC—quaternary ammonium compounds; OA—oxidizing agents; ChC—chlorine compounds; IC—iodine compounds; NANO—nanoparticles) was carried out, marking the MBC values. Based on the obtained results, the effectiveness coefficient (A) were calculated. The smaller the A value, the greater the efficiency of disinfection solutions prepared on the basis of ozonated versus non-ozonated water. Ozonated water showed biocidal efficacy against *L. monocytogenes*. Among tested disinfectentants, independent on type of water used for preparation, the most effective against *L. monocytogenes* were: QAC 1 (benzyl-C12-18-alkydimethyl ammonium chlorides) (1.00 × 10^−5^–1.00 × 10^−4^ g/mL) in quaternary ammonium compounds, OA 3 (peracetic acid, hydrogen peroxide, bis (sulphate) bis (peroxymonosulfate)) (3.08 × 10^−4^ –3.70 × 10^−3^ g/mL) in oxidizing agents, ChC 1 (chlorine dioxide) (5.00 × 10^−8^ –7.00 × 10^−7^ g/mL) in chlorine compounds, IC 1 (iodine) (1.05–2.15 g/mL) in iodine compounds, and NANO 1 (nanocopper) (1.08 × 10^−4^ – 1.47 × 10^−4^ g/mL) in nanoparticles. The values of the activity coefficient for QAC ranged from 0.10 to 0.40, for OA—0.15–0.84, for ChC—0.25–0.83, for IC—0.45–0.60, and for NANO—0.70–0.84. The preparation of disinfectants solution on the basis of ozonated water, improved the microbicidal efficiency of the tested disinfectant, especially the quaternary ammonium compounds. An innovative element of our work is the use of ozonated water for the preparation of working solutions of the disinfection agents. Use ozonated water can help to reduce the use of disinfectant concentrations and limit the increasing of microbial resistance to disinfectants. This paper provides many new information to optimize hygiene plans in food processing plants and limit the spread of microorganisms such as *L. monocytogenes*.

## 1. Introduction

*Listeria monocytogenes* causes listeriosis. This intracellular pathogen is widespread in the environment, from where it can enter the digestive tract of animals and humans [1]. The main source of *L. monocytogenes* is food, especially smoked fish, cheese, delicatessen meat products, milk, seafood, eggs, and vegetables [2].

Since *L. monocytogenes* is able to survive under food processing conditions it constitutes a serious threat in food processing plants. To prevent the spread of infection caused by this pathogen, chemical disinfection processes using compounds such as chlorine, iodine, oxidizing, phenolic, quaternary ammonium compounds, alcohols, aldehydes, or metal nanoparticles are carried out [3].

Recently ozone has become an alternative disinfectant. Because of its antibacterial properties, ozone is widely used for the disinfection of drinking water and sewage as well as in the food industry. In the disinfection processes, ozone is used in gaseous or aqueous form depending on the type of decontaminated surfaces. Low concentrations of ozone and short duration of action are sufficient to eliminate microorganisms [4]. It is active against bacteria (such as *Listeria* spp., *Escherichia* spp., *Salmonella* spp.), viruses, fungus, fungal spores, and protozoa [5]. Also, the constant ozonation of water with low ozone concentration (0.5 mg/L) intended for washing vegetables has resulted in a reduction in the number of mesophilic and coliform bacteria on the surface of lettuce and peppers. This method was less dense in relation to the elimination of mold and fungi. Also the type of vegetable plays a role in the effectiveness of the method (more effective for peppers) [6]. Ozone disturbs the integrity of the bacterial cell membrane by oxidizing phospholipids and lipoproteins. In the case of fungi, ozone inhibits microbial growth in a certain phase. In the case of viruses, ozone damages the viral capsid and interferes with the viral replication cycle [7]. Thanomsub et al. [8], using scanning electron microscopy (SEM), showed deformation of Gram-negative cells exposed to ozone at a concentration of 0.167/mg/min/L. After 60 min exposure, the cells were sunken and concave, while after 90 min of exposure, they showed lysis. The use of ozone does not require high temperatures and carries many economic benefits [9]. However, its short half-life period is associated with the need to produce it at the place of use. The half-life of ozone in an aqueous solution at 20 °C is approximately 20–30 min [10]. The use of ozonated water in the food industry determines the inclusion of organic matter and pH values. Arayan et al. [11] showed that organic pollution affects the ozone’s water half-life, and temperature is also an important factor. An increase in the temperature of ozone water caused a decrease in biocidal effectiveness [11]. Moreover ozone may be corrosive for the treated surface [12]. On the other hand its usage reduces the amount of other disinfectants, and herby the amount of their toxic by-products [4].

The aim of the study was to compare the biocidal effectiveness against *L. monocytogenes* strains of thirteen selected disinfectants, for which solutions were prepared using sterile hard water and ozone water. The aim was also to assess the stability of solutions of three disinfectants which effectiveness was significantly higher in ozonated water compared to hard water.

## 2. Materials and Methods

### 2.1. Bacterial Strains

The study was conducted on 6 *L. monocytogenes* strains isolated in 2015 from the territory of the Kuyavian-Pomeranian Voivodeship (Poland), including four isolates from food: meat (LMO-M), dairy products (LMO-N), vegetables (LMO-W), and fish (LMO-R), one clinical strain (LMO-K) from the collection of the Department of Microbiology, Nicolaus Copernicus University in Toruń, L. Rydygier Collegium Medicum in Bydgoszcz. These strains were susceptible to all antibiotics tested (penicillin, ampicillin, meropenem, erythromycin, cotrimoxazole) in accordance with the EUCAST v.8.00 recommendations [13]. The study material also included the reference strain *L. monocytogenes* ATCC 19111.

### 2.2. Ozonated Water

In the experiment, the hard water was used, prepared according to the Polish Standard PN-EN 1276: 2010 [14]. A solution A (19.84 g MgCl_2_ (Avantor, Gliwice, Poland) and 46.24 g CaCl_2_ (Avantor, Gliwice, Poland) was dissolved in 1000 mL H_2_O) and B (35.02 g NaHCO_3_ (Avantor, Gliwice, Poland) dissolved in 1000 mL H_2_O) was prepared to obtain a hard water. Both solutions were sterilized. Then, 6 mL of solution A and 8 mL of solution B were added to 700 mL of sterile water, mixed thoroughly and made up with sterile water to 1000 mL. The ozonation process of 100 mL of sterile hard water (temperature 20 °C, pH = 7.0) was carried out using a 20W ZE-H103 Orientee ozonator (ELTOM, Warsaw, Poland) with a diffuser and with the function of water and air ionization for 45 min. Ozone was generated in a laminar chamber at constant humidity and air temperature. Ozone concentration was determined in the reaction mixtures using DPD (*N*,*N*-diethyl-1,4-phenylene diammonium sulfate) Method, according ISO 7393-2:2017 [15], cuvette test and DR3900 Benchtop VIS Spectrophotometer provided by Hach (Frederick, Maryland, USA). This procedure of ozone determination was designed for water samples by Hach company. According to the procedure, the samples are treated with oxidizing agent *N*,*N*-diethyl-1,4-phenylene diammonium sulfate (DPD) to form a red dye. It was determined by visible spectrophotometry (λmax = 552 nm).

Due to the short typical half-life time of ozone in water (15–25 min for pH = 7–10) [16], ozonated water was immediately used to prepare solutions.

### 2.3. Disinfectants

The study used 13 disinfectants. The composition and concentration (in accordance with the manufacturer’s instructions) of working solutions needed to prepare the dilution series are presented in Table 1.

Taking into account the information provided by the manufacturer of a particular disinfector for the preparation of a commercial working solution (100%), the following solutions were prepared: 200%, 180%, 160%, 140%, 120%, 100%, 80%, 60%, 40%, 20%, 10%, 2%, and 1% of working solutions concentrations. The concentrations were selected in order to carry out the procedure for assessing the minimum bactericidal concentrations described in the further part of the methodology. The specific concentrations for particular tested disinfectants were presented in Table 2. Two independent dilution series were prepared—one using sterile hard water [14], the other—sterile ozonated water, immediately after its preparation.

### 2.4. Preparation of Bacterial Supensions

From cultures of *L. monocytogenes* strains obtained on Columbia Agar with 5% sheep blood (CAB, bioMérieux, Marcy-l’Étoile, France) suspensions of a density of 0.5 MacFarland standard (5.80 × 10^8^ CFU × mL^−1^) were prepared in 3 mL of Mueller Hinton Broth (MHB, Becton Dickinson, Franklin Lakes, New Jersey, USA). For this purpose, the optical density for the sterile MHB (Mueller Hinton Broth, Becton Dickinson, Franklin Lakes, New Jersey, USA) medium was first established. A sterile swab was then collected from a single colony grown on Columbia Agar with 5% sheep blood (CAB, bioMérieux, Marcy-l’Étoile, France) and loaded into the MHB (Mueller Hinton Broth, Becton Dickinson, Franklin Lakes, New Jersey, USA) medium, followed by measurement of the optical density of the suspension and subsequent colonization of *L. monocytogenes* added if necessary. The optical density of the suspension was set at 0.5 + the optical density of the sterile MHB (Mueller Hinton Broth, Becton Dickinson, Franklin Lakes, New Jersey, USA). The measurements were made with a DEN-1B denitometer from Biogenet (Józefów, Poland).

### 2.5. Assessment of Biocidal Effectiveness of Ozonated and Non-Ozonated Water

The suspensions of the tested *L. monocytogenes* strains (100 μL) were pipetted into Eppendorf (1.5 mL, Genoplast, Poland) tubes. After centrifugation (3000 rpm per 5 min) of the bacterial suspensions, 150 μL of an ozone solution of appropriate concentration (the mean value determined according to point 2.2–2.32 μg O_3_/mL) was added to the sediments. Immediately after preparation of the suspension in ozonated/non-ozonated water, a row of decimal dilutions were made in sterile PBS (Phosphate-buffered saline, Avantor, Gliwice, Poland). Each dilution (100 µL) was seeded into a Columbia Agar with 5% sheep blood (CAB, bioMérieux, Marcy-l’Étoile, France) and incubated for 24 h at 37°C. In this way, the initial number of *L. monocytogenes* was determined. The number of bacteria in the obtained suspension was 3.60–4.20 × 10^8^ CFU × mL^−1^. The negative control was 150 μL of ozon water, and the positive control—150 μL of bacterial suspension of a given strain suspended in sterile hard water. The study was carried out in triplicate for each strain and each tested concentration.

After 5 min of treatment of the suspensions with ozonated water, sample were transferred to 900 μL neutralizer (10 g Tween 80 (Sigma Aldrich, Saint Louis, Missouri, USA), 1 g lecithin (Sigma Aldrich, Saint Louis, Missouri, USA), 0.5 g histidine L (Sigma Aldrich, Saint Louis, Missouri, USA), 2.5 g Na_2_S_2_O_3_ (Avantor, Gliwice, Poland), 3.5 g C_3_H_3_NaO_3_ (Avantor, Gliwice, Poland), 1000 mL sterile water). Lecithin neutralizes quaternary ammonia compounds while phenolic disinfectants and hexachlorophene are neutralized by Tween. Together, lecithin and Tween neutralize ethanol. Histidine inactivates aldehydes, especially formaldehyde and gluteraldehyde. Sodium thiosulfate neutralizes iodine and chlorine, whereas sodium pyruvate neutralizes active oxygen and peroxides [17,18]. After 5 min of exposure, linear cultures were made on the CAB (Columbia Agar with 5% sheep blood, bioMérieux, Marcy-l’Étoile, France) substrate plate sectors that were incubated for 24 h at 37 °C. After incubation, the concentration of ozonated water was analyzed, which enabled the inactivation of the tested strains of *L. monocytogenes*. The effect of sterile hard water [15] on the number of recovered bacteria was also assessed.

After determining the concentration range in which the value of the minimum bactericidal concentration (MBC) of ozonated water was located, the procedure was repeated, preparing solutions with a concentration varying by 1% in this range, in order to accurately determine the MBC (minimum bactericidal concentration).

To check the durability of ozonated water, the same test cycle was carried out one and two h after the ozonation process was completed. All plates with banded cultures were incubated under the conditions described above, and then the results were read.

### 2.6. Evaluation of the Effectiveness of Disinfectants

The suspensions of the tested strains of *L. monocytogenes* (100 μL) and 100 μL of the appropriate concentration of disinfectant were introduced into the wells of a multi-well polystyrene plate. The target concentration of the disinfectant in the well plate was respectively 100%, 90%, 80%, 70%, 60%, 50%, 40%, 30%, 20%, 10%, 5%, 1%, and 0.5% concentration working solution of a particular disinfectant. The specific concentrations for particular tested disinfectants were presented in Table 2. The negative control consisted of 200 μL of sterile MHB (Mueller Hinton Broth, Becton Dickinson, Franklin Lakes, New Jersey, USA) medium, and a positive control—200 μL of bacterial suspension. After 5 min of the agent’s action on bacterial suspensions, 100 μL of liquid was transferred from each well to 900 μL of neutralizer. After 5 min of neutralization, band-cultures were made on the CAB (Columbia Agar with 5% sheep blood, bioMérieux, Marcy-l’Étoile, France) segments, which were incubated for 24 h at 37°C. After incubation, the minimum concentration allowing inactivation of the tested strains of *L. monocytogenes* was read for solutions based on non-ozonated and ozonated water.

After determining the concentration range in which the value of the minimum bactericidal concentration (MBC) of a given disinfectant was located, the procedure was repeated by preparing solutions with a concentration varying by 1% in this range, to accurately determine the MBC.

Based on the obtained results, for all strains and disinfectants used in the studies, the effectiveness coefficient (A) were calculated [19]. The smaller the value of the coefficient A, the greater the efficiency of disinfecting solutions based on ozonated water compared to non-ozonated disinfectants.

The effectiveness factor was calculated from the formula:A = b/c
where:

A—effectiveness coefficient,

b—effective concentration of disinfectant in active solution (ozonated water),

c—effective concentration of disinfectant in non-ozonated water solution.

In order to assess the decrease in the number of *L*. *monocytogenes* under the influence of disinfectants prepared on the basis of ozonated and non-ozonated water, the initial number of tested bacteria and the number of bacteria isolated at preMBC disinfectant concentration were determined. For this purpose, a series of decimal dilutions was made for the prepared suspension of a given strain and then plated on CAB (Columbia Agar with 5% sheep blood, bioMérieux, Marcy-l’Étoile, France) agar. The cultures were incubated at 37°C for 24 h. After this time, the grown colonies were counted and converted into logarithmic units. Similarly, the suspension subjected to action of the disinfectant concentration directly preceding MBC (preMBC) was treated. To determine preMBC, the MBC value from Table 3 was checked and the directly preceding concentration was selected from Table 2. The decreases in the number of bacteria were calculated from the formula:R = log_i_ − log_preMBC_
where:

R—reduction in bacteria number (log CFU)

log_i_—initial bacteria number

log_preMBC_—nuber of bacteria recovered from suspension trated with preMBC concentration of disinfectant

### 2.7. Assessment of the Stability of QAC 2, OA 3, and ChC1 Solutions

Three disinfectants, for which the effectiveness coefficient was the lowest, were evaluated for the stability (QAC 2, OA 3, and ChC 1). The prepared working solutions, both in non-ozonated and ozonated water, were stored at room temperature. The effectiveness of these agents on *L. monocytogenes* strains was evaluated immediately after preparation of the solutions, after 12 and 24 h, determining the MBC values in individual time intervals. This part of the experiment was designed to assess whether the possible increased effectiveness of disinfectants is not prolonged due to some reactions between ozone and active substances. On the basis of MBC values, the coefficient A was calculated for particular disinfectant, strains and time of storage.

### 2.8. Statistical Analysis

Statistical analysis was performed in the STATISTICA 13.1 PL program (StatSoft). Significance of differences between the ozonated water effectiveness, MBC values of disinfectants, reduction in *L. monocytogenes* number and maximal effectiveness coefficients for disinfectants were assessed.

#### 2.8.1. Biocidal Effectiveness of Ozonated and Non-Ozonated Water

It was checked how the MBC value of the ozonated water varied depending on the time (0, 1, and 2 h) and the *L*. *monocytogenes* strain. Experiment was made in 3 replications. As independent variables, the strain as well as the time elapsed from the ozonation of water were treated, and as a dependent variable the determined value of MBC was recognized. Significance of differences between mean MBC values for the combination of both independent variables was checked. For this purpose, the general line models (GLM) were used. The multi-way ANOVA was conducted. The Tukey post-hoc test was used for significance of *α* = 0.05.

#### 2.8.2. Evaluation of the Effectiveness of Disinfectants

It was checked how the MBC value of tested disinfectants varied depending on the type of water (ozonated/non-ozonated) used for solutions preparation. Experiment was made in 3 replications. As independent variables, the disinfectant, type of water, as well as the strain were treated, and as a dependent variable the determined value of disinfectants MBC was recognized. Significance of differences between mean MBC values for the combination of all independent variables was checked. For this purpose, the general line models (GLM) were used. The multi-way ANOVA was conducted. The Tukey post-hoc test was used for significance of *α* = 0.05.

#### 2.8.3. Reduction in Bacteria Number

It was checked how the reduction in bacteria number varied depending on the type of water (ozonated/non-ozonated) used for disinfecting solutions preparation. Experiment was made in 3 replications. As independent variables, the type of water was treated, and as a dependent variable the determined reduction in bacteria number was recognized. Significance of differences between mean reduction obtained for particular disinfectant depending on type of water was checked. For this purpose, the general line models (GLM) were used. The one-way ANOVA was conducted. The Tukey post-hoc test was used for significance of *α* = 0.05.

#### 2.8.4. Maximal Effectiveness Coefficients for Disinfectants

It was checked how the maximal effectiveness coefficients for disinfectants varied depending on the type of water (ozonated/non-ozonated) used for disinfecting solutions preparation. Experiment was made in 3 replications. As independent variables, the type of water was treated, and as a dependent variable the determined maximal effectiveness coefficients for disinfectants was recognized. Significance of differences between mean coefficient value obtained for particular disinfectant depending on type of water was checked. For this purpose, the general line models (GLM) were used. The one-way ANOVA was conducted. The Tukey post-hoc test was used for significance of *α* = 0.05.

## 3. Results

### 3.1. Assessment of Biocidal Effectiveness of Ozonated and Non-Ozonated Water

The non-ozonated water did not show biocidal efficacy against the tested strains of *L. monocytogenes*.

The determined concentration of ozone in ozonated water was 2.32 μg/mL (± 0.022 μg/mL). The ozonated water, used immediately after preparation, inhibited bacterial growth at ozone concentrations 1.86–1.96 μg/mL, with the lowest resistance characterized for the reference strain (1.86 μg/mL) and the highest—for strains derived from meat and fish (1.96 μg/mL). For the ozonated water after two h from the end of the ozonation process, for all tested *L. monocytogenes* strains, the total lack of efficacy was demonstrated (Figure 1).

### 3.2. Evaluation of Effectiveness of Disinfectant

Disinfectants based on ozonated water were characterized by a higher biocidal efficiency than solutions based on non-ozonated water (Table 3). In most cases these differences were statistically significant (*p* ≤ 0.05) (Table 3).

Among the tested disinfectants, regardless of type of water used for preparation, the most effective against *L. monocytogenes* were: QAC 1 (1.00 × 10^−5^ –1.00 × 10^−4^ g/mL) in quaternary ammonium compounds, OA 3 (3.08 × 10^−4^–3.70 × 10^−3^ g/mL) in oxidizing agents, ChC 1 (5.00 × 10^−8^ –7.00 × 10^−7^ g/mL) in chlorine compounds, IC 1 (1.05–2.15 g/mL) in iodine compounds and NANO 1 (1.08 × 10^−4^ –1.47 × 10^−4^ g/mL) in nanoparticles (Table 3). In case of NANO 2, the solution based on non-ozonated water, was totally ineffective against all tested *L. monocytogenes* strains. Moreover, the OA 2 solution based on non-ozonated water was also ineffective against LMO-W, LMO-M and LMO-R strains (Table 3).

The MBC value of tested disinfectants determined for particular examined strains of *L. monocytogenes* were very similar, so the effect was not strain-dependent. None strain-dependent differences in MBC were stated in case of QAC 2, QAC 3, OA 1, OA 4 and NANO 2 (Table 3). In all cases the concentration of disinfectants equal MBC value cause decrease in bacteria number below the detection limit. In Table 4, the logarithmic decreases in bacteria number after using the disinfectant concentration directly preceding MBC (preMBC), for solution based on ozonated and non-ozonated water, are presented. For solutions of disinfectants prepared on the basis of ozonated water, the determined preMBC values were lower, and despite this the logarithmic decreases in the number of *L*. *monocytogenes* were found to be higher. The observed differences in decreases were, in most cases, statistically significant (*p* ≤ 0.05) (Table 4).

### 3.3. Coefficients of Effectiveness of Disinfectants

The values of the activity coefficient for all strains and disinfectants are shown in Table 5. These values for quaternary ammonium compounds ranged from 0.10 to 0.40, for oxidizing agents—from 0.15 to 0.84, for chlorine compounds—from 0.25 to 0.83, for iodine compounds—from 0.45 to 0.60 and for nanoparticles—from 0.70 to 0.84 (Table 5). It was shown, ozonated water had the greatest impact on the efficiency of the quaternary ammonium compounds whereas did not significantly improve the effectiveness of nanoparticles (Figure 2).

It was not possible to select the *L. monocytogenes* strain, for which the increase in the effectiveness of all tested disinfectants would be the greatest or the smallest after preparation of solutions with ozonated water (Table 5).

The lowest maximal value of the efficacy coefficient, and therefore the highest increase in microbicidal effectiveness after using ozonated water, was demonstrated for QAC 2 (0.10) and QAC 3 (0.20) and OA 3 (0.23). The values of the efficacy coefficient differ significantly (*p* > 0.05) between QAC 2 versus QAC 3 and OA 3 (Figure 2). The highest maximal efficacy coefficients, indicating a small improvement in the bactericidal effectiveness of solutions based on ozonated water in relation to solutions prepared with non-ozonated water, were found for OA 1 (0.84), ChC 2 (0.83) and NANO 1 (0.84). The above values of coefficients differed statistically significantly (*p* ≤ 0.05) with the majority of values calculated for the remaining disinfectants (Figure 2).

### 3.4. Assessment of the Stability of QAC 2, OA 3, and ChC 1 Solutions

QAC 2, OA 3, and ChC 1 were chosen as the agents from different groups with the lowest maximal value of efficacy to assess the stability of the solutions. The stability results of the tested solutions are shown in Table 6.

It has been shown that during the storage of disinfectant solutions, their biocidal activity decreases against *L. monocytogenes* strains. Decrease in the biocidal activity of disinfectant solutions, which were prepared using both ozonated and non-ozonated water, was observed. The greater decrease in biocidal activity was visible for solutions prepared with the use of ozonated water (Table 6).

The activity of the QAC 2 solutions decreased over all tested strains after 24 h of storage, both for solutions with non-ozonated and ozonated water. In the case of OA 3 and ChC 1 disinfectants, a decrease in activity was observed for all tested strains of *L. monocytogenes* after 12 and 24 h of storage, for both type of water used for solutions preparation (Table 6).

For all the disinfectants included in this part of the study, a gradual increase in the efficiency coefficient was observed during storage (Table 7). For OA 3 and ChC 1, after 12 h from preparation, the antilisterial effectiveness of the solution based on ozonated and non-ozonated water was almost identical. For QAC 2, this was observed after 24 h (Table 7). After 24 h, in the case of QAC 2 for LMO-R strain, OA 3 strains for LMO-W and LMO-N strains and ChC 1 for LMO-N strain, the effectiveness of solution based on ozonated water was lower than for those prepared on non-ozonated water (Table 7).

## 4. Discussion

Ozone is one of the strongest disinfectants, which are active after a short contact time and in low concentration. It has biocidal effect on many types of microorganisms [4]. Muthukumar and Muthuchama’s [20] studies confirmed a decrease in the number of microorganisms by 2.00 × 10^6^ CFU in 1g of raw chicken samples. While Sheelamary and Muthukumar [21] showed complete inactivation of microorganisms isolated from samples of milk and its products after only 15 min with emission of 0.2 g O_3_/h.

In the literature, research work on the evaluation of the effectiveness of gas ozone in combination with various physical methods has been found [22,23,24,25]. Sung et al. [23] examined the action of gas ozone and high temperature on inactivation of *L. monocytogenes* present in apple juice. The synergistic effect was obtained when the temperature was 50 °C. Kumar et al. [25] during a 10-min exposure to ozone gas and UV showed a decrease in the number of *L. monocytogenes* in fresh brine by more than 9 log CFU/mL, and hourly ozonation combined with ten-minute UV irradiation resulted in a reduction of microbes over 5 log CFU/mL in used brines. The effect of low concentrations of ozone and metal ions in the reduction of *L. monocytogenes* was studied by Kang et al. [26], who showed that the use of ozone in concentrations of 0.2 and 0.4 ppm in combination with 1 mM CuCl_2_ and 0.1 mM AgNO_3_ by 30 min is much more effective than using ozonated water (*p* < 0.05) only. Marino et al. [27] evaluated the effect of gas ozone and ozone in water on the survival of *Pseudomonas fluorescens*, *Staphylococcus aureus,* and *L. monocytogenes* cells in the biofilm structure on the surface of stainless steel. They showed that the use of ozone in the aqueous solution affected the reduction of the number of bacteria by 1.61–2.14 log CFU/cm^2^ after 20 min exposure, while the reduction values were higher (3.26–5.23 log CFU/cm^2^) in the case of biofilms treated with ozone under dynamic flow conditions. They also showed that *S. aureus* was the most sensitive species for ozone dissolved in water [27]. Korany et al. [28] showed that the use of ozonated water (1 min) against the structure of *L. monocytogenes* biofilm on the surface of polystyrene at the concentration of 1.0, 2.0, and 4.0 ppm resulted in a bacterial number reduction of 0.9, 3.4, and 4.1 log CFU/cm^2^, respectively. Moreover Korany et al. [28] found that quaternary ammonium compounds (QAC) (100/400 ppm), chlorine (100/200 ppm), chlorine dioxide (2.5/5.0 ppm) and peracetic acid (PAA) (80/160 ppm) resulted in a reduction of 2.4/3.6, 2.0/3.1, 2.4/3.8, and 3.6/4.8 log CFU/cm^2^, respectively. The antimicrobial efficacy of all tested disinfectants against the 7-day *L. monocytogenes* biofilm was significantly lower compared to 2-day biofilms, and the biofilm age having the minor impact on the effectiveness of PAA [28].

To date, information on the synergistic effect of ozonated water and disinfectants was not found. However, it should be expected that the implementation of a solution of disinfectants based on ozonated water will increase the effectiveness of the tested agents, at least up to the level of additive action.

In this study, the biocidal efficacy of ozonated water and non-ozonated water was evaluated. Non-ozonated water was characterized by a complete lack of biocidal activity, against all tested strains of *L. monocytogenes*. The ozonated water, used immediately after preparation, containing 1.86–1.96 μg O_3_/cm^3^, was effective for all tested isolates. The LMO-M and LMO-R strains were characterized by the highest resistance (MBC: 1.96 μg/cm^3^), and the LMO-ATCC strain (MBC: 1.86 μg/cm^3^) was the most sensitive. Fishburn et al. [29] showed that the use of ozonated water in vegetable washing can cause a decrease in the number of *L. monocytogenes* strains by about 0.5 log CFU/g (broccoli, lettuce) to 1.5 log CFU/g (green onion), in relation to washing in non-ozonated water. Larivière-Gauthier et al. [30] have shown that the use of ozonated water containing 3.5 ppm ozone improves the efficiency of cleaning and disinfection in a pork cutting plant, increasing the proportion of free surfaces from *L. monocytogenes* by 12.5%. Arayan et al. [11] showed that the lowest biocidal concentration against *Staphylococcus aureus* (strains isolated from food products) was 0.5 ppm of ozone with an exposure time of 0.1 min. The addition of organic pollutants such as fetal bovine serum (FBS) resulted in a decrease in the biocidal effectiveness of the ozonated water [11]. Also Korany et al. [28] showed lower efficacy of disinfectants, including ozonated water, in case of the presence of organic contamination (diluted milk and apple juice) in the environment.

The results of our study showed that after 1 and 2 h of storage of ozonated water its activity decreased, as evidenced by the need to use higher concentrations of ozone to eliminate *L. monocytogenes* strains after longer storage period of ozonated water. This was probably related to the decomposition of ozone, which is unstable and has a water stability of 20–40 min. This is confirmed by the studies of Białoszewski et al. [12] who used ozonated water after 30 min of its preparation and observed a decrease in ozone concentration from the initial 2.5–3.0 μg/mL to 1.3–1.5 μg/mL. Despite the decrease in ozone content, they did not show a decrease in the biocidal effectiveness of water. Research carried out by Seki et al. [10] showed that storage of ozonated water at 4 °C and 25 °C significantly affected its durability. The highest decrease (90.0%) in ozone concentration was noticed after one week of storage at 25 °C. However, tests carried out at 4 °C by Seki et al. [10] showed that after one week the concentration of ozone in water maintained at the level of above 90%, after a month about 65%, and after one year of storage ozone was not found in water. The effect of biocidal ozonated water stored at 4°C against *Escherichia coli* has been confirmed. Seki et al. [10] also assessed the effect of freezing and thawing on the durability of ozone in water. They showed that the first freeze-thaw cycle did not affect the ozone concentration in water, but after four cycles it was found that the ozone concentration decreased to about 90% [10].

This study allowed to assess and compare the biocidal efficacy of solutions of thirteen selected disinfectants made using non-ozonated water and ozonated water against *L. monocytogenes* strains. It was shown that disinfectants based on ozonated water were characterized by higher microbicidal effectiveness, compared to solutions based on non-ozonated water and were more effective than ozonated water alone. This shows the synergistic effect of ozonated water and disinfectants. This may be related to the same target site of the test compound and ozone in bacterial cells. For example, QACs or peracetic acid act similarly to ozone, destabilizing the bacterial cell membrane, making it more permeable [31,32]. The synergistic mechanism of action can also be the result of influence of ozone and the active substance of the disinfectants on other structures of the bacterial cell, which will result in an increase of microbicidal effect. Oxidizing compounds (e.g., hydrogen peroxide, hypochlorous acid, and peracetic acid) cause the oxidation of thiol groups of cysteine residues, which are often found in the active sites of many bacterial enzymes, such as, for example, dehydrogenases.

It was found that the use of ozonated water for solution preparation, improves the microbicidal properties of all tested disinfectants, with the exception of nanosilver. The greatest impact of ozonated water on disinfectant effectiveness was noted for quaternary ammonium compounds. This type of disinfectants displayed the highest biocidal efficacy against tested isolates. The strong biocidal activity of quaternary ammonium compounds against *L. monocytogenes* strains is confirmed by the results of Chavant et al. [33], which showed 98% effectiveness in eliminating planktonic forms. In contrast, Aarnisalo et al. [34] have shown that quaternary ammonium compounds are characterized by lower biocidal efficacy against *L. monocytogenes* strains than disinfectants based on chlorine, ethanol, isopropanol and peracetic acid. In our study, the LMO-R strain was the most resistant to the effect of NANO 1, OA 2, and OA 3 preparations belonging to oxidizing compounds, and the LMO-M strain was the most resistant to the ChC 1 compound from the group of chlorine derivatives. The LMO-ATCC strain was characterized by the highest sensitivity against ChC 1. Heir et al. [35] and Popowska et al. [36] showed different sensitivity of *L. monocytogenes* to disinfectants depending on the origin of the strain.

The conducted studies also evaluated the durability of solutions with the lowest efficiency index for QAC 2, OA 3, and ChC 1. The decrease in the biocidal activity of disinfectant solutions prepared using non-ozonated water may suggest that they are gradually inactivated during storage.

The results of our study show that ozonated water have microbicidal properties. Since ozonated water and disinfectants show a synergistic effect of a biocidal action its combination can be used to more effectively eradicate microorganisms. Moreover ozonated water can reduce the working concentration of disinfectants. However, it is important to remember about the short half-life of ozone and to use it directly or in a short time after the preparation of appropriate disinfecting solutions.

To elucidate the mechanism of joint action of disinfectants and ozonated water on the disruption of bacterial cell structure further research are needed. In our study, the Columbia Agar with 5% sheep blood was used, which is a non-selective medium. The use of such a medium provided suitable conditions for the regeneration of sub-lethal injured cells. Presence of selective and differential agents in a medium could inhibit the growth of such cells relative to non-selective medium and thus counts obtained from treated samples enumerated on selective media are typically lower than those obtained from non-selective media. This tendency is confirmed by studies by Fouladkhah et al. [37] who showed that when assessing the growth of bacteria on selective medium on day 0 the number of colonies ranged from 1.5 ± 0.8 to 2.0 ± 0.8 log CFU/cm^2^ and increased to values from 2.9 ± 0.5 to 4.3 ± 0.4 log CFU/cm^2^ on 7 day. In turn, the results obtained after cultivation on non-selective media ranged from 2.0 ± 0.5 to 2.3 ± 0.4 log CFU/cm^2^ on day 0 and from 6.4 ± 0.6 to 7.1 ± 0.4 log CFU/cm^2^ on day 7 [37].

## 5. Conclusions

We can state that the ozonated water, in contrast to non-ozonated water, showed biocidal efficacy against examined *L. monocytogenes* strains though its activity decreases over time. Moreover, it was demonstrated that ozonated water improved the biocidal effectiveness of disinfecting agents. Among the tested solutions of disinfectant, the most effective group were quaternary ammonium compounds and chlorine compounds. The lowest biocidal activity against the tested strains of *L. monocytogenes* were characterized by nanoparticles. The biocidal activity of disinfectants against tested *L. monocytogenes* strains decreases during storage regardless of the disinfectant type.

## Figures and Tables

**Figure 1 microorganisms-07-00127-f001:**
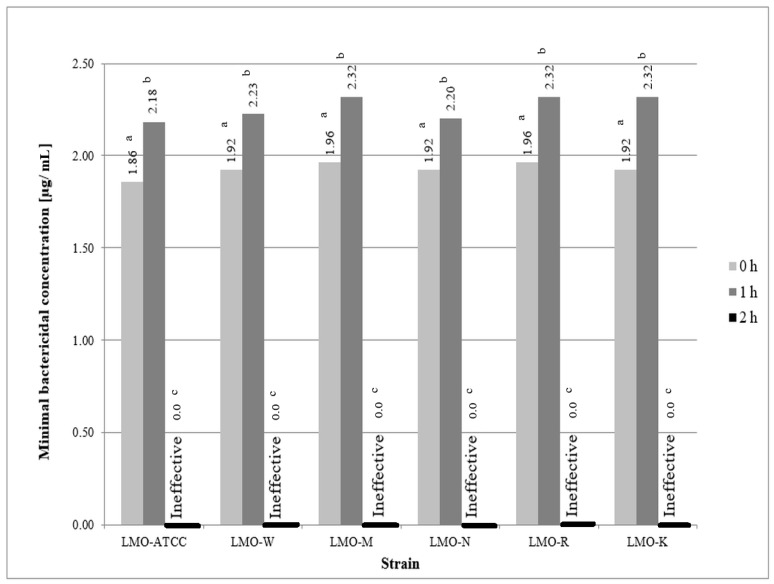
Effectiveness of ozonated water against the tested strains of *L. monocytogenes (*LMO-ATCC—*L. monocytogenes* ATCC 19111, LMO-W—strain isolated from vegetables, LMO-M—strain isolated from meat, LMO-N—strain isolated from dairy products, LMO-R—strain isolated from fish, LMO-K—clinical strain.; a,b,c—variables with different letters are statistically different (*p* ≤ 0.05).

**Figure 2 microorganisms-07-00127-f002:**
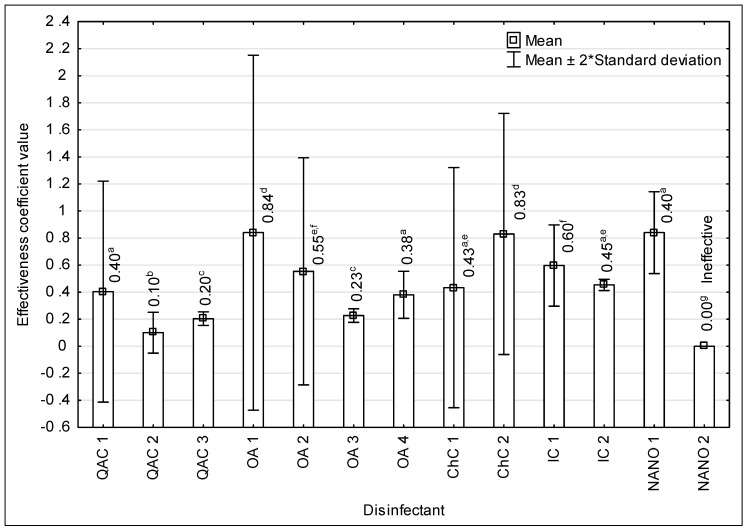
Values of efficacy coefficients for the tested disinfectants, including their belonging to distinguished groups (QAC—quaternary ammonium compounds, OA—oxidizing agents, ChC—chlorine compounds, IC—iodine compounds, NANO—nanoparticles); a–g—variables with different letters are statistically different (*p* ≤ 0.05)

**Table 1 microorganisms-07-00127-t001:** Characteristics of disinfectants.

Group of Disinfectants	Name	Active Substance	Working Solution Concentration (g/mL)
Quaternary ammonium compounds	QAC 1	benzyl-C12-18-alkydimethyl ammonium chlorides	2.0 × 10^−3^
	QAC 2	benzyl-C12-16 alkyldimethyl chlorides	2.55 × 10^0^
	QAC 3	didecyldimethylammonium chloride, benzyl-C12-16-alkyldimethyl chlorides	2.97 × 10^0^
Oxidizing agents	OA 1	hydrogen peroxide, silver nitrate	1.20 × 10^1^
	OA 2	perlactic acid	4.90 × 10^0^
	OA 3	peracetic acid, hydrogen peroxide	6.15 × 10^−3^
	OA 4	bis (sulphate) bis (peroxymonosulfate) pentapotassium, benzenesulfonic acid, C10-13 alkyl derivatives, sodium salts, malic acid, sulfamic acid	2.00 × 10^−2^
Chlorine compounds	ChC 1	chlorine dioxide	1.00 × 10^−5^
	ChC 2	hypochlorous acid calcium salt	2.00 × 10^−3^
Iodine compounds	IC 1	iodine	6.15 × 10^0^
	IC 2	iodine	1.17 × 10^1^
Nanoparticles	NANO 1	nanocopper	1.50 × 10^−4^
	NANO 2	nanosilver	1.50 × 10^−4^

**Table 2 microorganisms-07-00127-t002:** The specific concentrations for particular tested disinfectants.

Group of Disinfectants	Name	Initial Concentration (g/mL)	Final Concentration (g/mL)
Quaternary ammonium compounds	QAC 1	4.00 × 10^−3^; 3.60 × 10^−3^; 3.20 × 10^−3^; 2.80 × 10^−3^; 2.40 × 10^−3^; 2.00 × 10^−3^; 1.60 × 10^−3^; 1.20 × 10^−3^; 8.00 × 10^−4^; 4.00 × 10^−4^; 2.00 × 10^−4^; 4.00 × 10^−5^; 2.00 × 10^−5^	2.00 × 10^−3^; 1.80 × 10^−3^; 1.60 × 10^−3^; 1.40 × 10^−3^; 1.20 × 10^−3^; 1.00 × 10^−3^; 8.00 × 10^−4^; 6.00 × 10^−4^; 4.00 × 10^−4^; 2.00 × 10^−4^; 1.00 × 10^−4^; 2.00 × 10^−5^; 1.00 × 10^−5^
	QAC 2	5.1 × 10^0^; 4.59 × 10^0^; 4.08 × 10^0^; 3.57 × 10^0^; 3.06 × 10^0^; 2.55 × 10^0^; 2.04 × 10^0^; 1.53 × 10^0^; 1.02 × 10^0^; 5.1 × 10^−1^; 2.6 × 10^−1^; 5.00 × 10^−2^; 2.6 × 10^−2^	2.55 × 10^0^; 2.30 × 10^0^; 2.04 × 10^0^; 1.79 × 10^0^; 1.53 × 10^0^; 1.28 × 10^0^; 1.02 × 10^0^; 7.70 × 10^−1^; 5.10 × 10^−1^; 2.60 × 10^−1^; 1.30 × 10^−1^; 2.60 × 10^−2^; 1.00 × 10^−2^
	QAC 3	5.94 × 10^0^; 5.35 × 10^0^; 4.75 × 10^0^; 4.16 × 10^0^; 3.56 × 10^0^; 2.97 × 10^0^; 2.38 × 10^0^; 1.78 × 10^0^; 1.19 × 10^0^; 5.90 × 10^−1^; 3.00 × 10^−1^; 5.90 × 10^−2^; 3.00 × 10^−2^	2.97 × 10^0^; 2.67 × 10^0^; 2.38 × 10^0^; 2.08 × 10^0^; 1.78 × 10^0^; 1.49 × 10^0^; 1.19 × 10^0^; 8.90 × 10^−1^; 5.90 × 10^−1^; 2.30 × 10^−1^; 1.50 × 10^−1^; 3.00 × 10^−2^; 1.50 × 10^−2^
Oxidizing agents	OA 1	2.40 × 10^1^; 2.16 × 10^1^; 1.92 × 10^1^; 1.68 × 10^1^; 1.44 × 10^1^; 1.20 × 10^1^, 9.60 × 10^0^; 7.20 × 10^0^; 4.80 × 10^0^; 2.40 × 10^0^; 1.20 × 10^0^; 2.40 × 10^−1^; 1.20 × 10^−1^	1.20 × 10^1^; 1.08 × 10^1^; 9.60 × 10^0^; 8.40 × 10^0^; 7.20 × 10^0^; 6.00 × 10^0^; 4.80 × 10^0^; 3.60 × 10^0^; 2.40 × 10^0^; 1.20 × 10^0^; 6.00 × 10^−1^; 1.20 × 10^−1^; 6.00 × 10^−2^
	OA 2	9.80 × 10^0^; 8.82 × 10^0^; 7.84 × 10^0^; 6.86 × 10^0^; 5.88 × 10^0^, 4.90 × 10^0^; 3.92 × 10^0^; 2.94 × 10^0^; 1.96 × 10^0^; 9.80 × 10^−1^; 4.90 × 10^−1^; 9.80 × 10^−2^; 4.90 × 10^−2^	4.90 × 10^0^; 4.41 × 10^0^; 3.92 × 10^0^; 3.43 × 10^0^; 2.45 × 10^0^; 2.40 × 10^0^; 1.96 × 10^0^; 1.47 × 10^0^; 9.80 × 10^−1^; 4.90 × 10^−1^; 2.50 × 10^−1^; 5.00 × 10^−2^; 2.50 × 10^−2^
	OA 3	1.20 × 10^−2^; 1.00 × 10^−2^; 9.80 × 10^−3^; 8.60 × 10^−3^; 7.40 × 10^−3^; 6.15 × 10^−3^; 4.92 × 10^−3^; 3.69 × 10^−3^; 2.46 × 10^−3^; 1.23 × 10^−3^; 6.15 × 10^−4^; 1.23 × 10^−4^; 6.15 × 10^−5^	6.15 × 10^−3^; 5.50 × 10^−3^; 4.90 × 10^−3^; 4.30 × 10^−3^; 3.70 × 10^−3^; 3.10 × 10^−3^; 2.46 × 10^−3^; 1.90 × 10^−3^; 1.23 × 10^−3^; 6.15 × 10^−4^; 3.10 × 10^−4^; 6.15 × 10^−5^; 3.10 × 10^−5^
	OA 4	4.00 × 10^−2^; 3.60 × 10^−2^; 3.20 × 10^−2^; 2.80 × 10^−2^; 2.40 × 10^−2^; 2.00 × 10^−2^; 1.60 × 10^−2^; 1.20 × 10^−2^; 8.00 × 10^−3^; 4.00 × 10^−3^; 2.00 × 10^−3^; 4.00 × 10^−4^; 2.00 × 10^−4^	2.00 × 10^−2^; 1.80 × 10^−2^; 1.60 × 10^−2^; 1.40 × 10^−2^; 1.20 × 10^−2^; 1.00 × 10^−2^; 8.00 × 10^−3^; 6.00 × 10^−3^; 4.00 × 10^−3^; 2.00 × 10^−3^; 1.00 × 10^−3^; 2.00 × 10^−4^; 1.00 × 10^−4^
Chlorine compounds	ChC 1	2.00 × 10^−5^; 1.8 × 10^−5^; 1.6 × 10^−5^; 1.4 × 10^−5^; 1.20 × 10^−5^; 1.00 × 10^−5^; 8.00 × 10^−6^; 6.00 × 10^−6^; 4.00 × 10^−6^; 2.00 × 10^−6^; 1.00 × 10^−6^; 2.00 × 10^−7^; 1.00 × 10^−7^	1.00 × 10^−5^; 9.00 × 10^−6^; 8.00 × 10^−6^; 7.00 × 10^−6^; 6.00 × 10^−6^;5.00 × 10^−6^; 4.00 × 10^−6^; 3.00 × 10^−6^; 2.00 × 10^−6^; 1.00 × 10^−6^; 2.00 × 10^−7^; 1.00 × 10^−7^
	ChC 2	4.00 × 10^−3^; 3.60 × 10^−3^; 3.20 × 10^−3^; 2.80 × 10^−3^; 2.40 × 10^−3^; 2.00 × 10^−3^; 1.60 × 10^−3^; 1.20 × 10^−3^; 8.00 × 10^−4^; 4.00 × 10^−4^; 2.00 × 10^−4^; 4.00 × 10^−5^; 2.00 × 10^−5^	2.00 × 10^−3^; 1.80 × 10^−3^; 1.60 × 10^−3^; 1.40 × 10^−3^; 1.20 × 10^−3^; 1.00 × 10^−3^; 8.00 × 10^−4^; 6.00 × 10^−4^; 4.00 × 10^−4^; 2.00 × 10^−4^; 1.00 × 10^−4^; 2.00 × 10^−5^; 1.00 × 10^−5^
Iodine compounds	IC 1	1.23 × 10^1^; 1.11 × 10^1^; 9.84 × 10^0^; 8.61 × 10^0^; 7.38 × 10^0^; 6.15 × 10^0^; 4.92 × 10^0^; 3.69 × 10^0^; 2.46 × 10^0^; 1.23 × 10^0^; 6.20 × 10^−1^; 1.20 × 10^−1^; 6.20 × 10^−2^	6.15 × 10^0^; 5.54 × 10^0^; 4.92 × 10^0^; 4.31 × 10^0^; 3.69 × 10^0^; 3.08 × 10^0^; 2.46 × 10^0^; 1.85 × 10^0^; 1.23 × 10^0^; 6.20 × 10^−1^; 3.10 × 10^−1^; 6.20 × 10^−2^; 3.10 × 10^−2^
	IC 2	2.34 × 10^1^; 2.11 × 10^1^; 1.87 × 10^1^; 1.64 × 10^1^; 1.40 × 10^1^; 1.17 × 10^1^; 9.36 × 10^0^; 7.02 × 10^0^; 4.68 × 10^0^; 2.34 × 10^0^; 1.17 × 10^0^; 2.30 × 10^−1^; 1.20 × 10^−1^	1.17 × 10^1^; 1.05 × 10^1^; 9.36 × 10^0^; 8.19 × 10^0^; 7.02 × 10^0^; 5.85 × 10^0^; 4.68 × 10^0^; 3.51 × 10^0^; 2.34 × 10^0^; 1.17 × 10^0^; 5.90 × 10^−1^; 1.20 × 10^−1^; 5.90 × 10^−2^
Nanoparticles	NANO 1	3.00 × 10^−3^; 2.70 × 10^−3^; 2.40 × 10^−3^; 2.10 × 10^−3^; 1.80 × 10^−3^; 1.50 × 10^−3^; 1.20 × 10^−3^; 9.00 × 10^−4^; 6.00 × 10^−4^; 3.00 × 10^−4^; 1.50 × 10^−4^; 3.00 × 10^−5^; 1.50 × 10^−5^	1.50 × 10^−4^; 1.35 × 10^−4^; 1.20 × 10^−4^; 1.05 × 10^−4^; 9.00 × 10^−5^; 7.50 × 10^−5^; 6.00 × 10^−5^; 4.50 × 10^−5^; 3.00 × 10^−5^; 1.50 × 10^−5^; 7.50 × 10^−6^; 1,50 × 10^−6^; 7.50 × 10^−7^
	NANO 2	3.00 × 10^−3^; 2.70 × 10^−3^; 2.40 × 10^−3^; 2.10 × 10^−3^; 1.80 × 10^−3^; 1.50 × 10^−3^; 1.20 × 10^−3^; 9.00 × 10^−4^; 6.00 × 10^−4^; 3.00 × 10^−4^; 1.50 × 10^−4^; 3.00 × 10^−5^; 1.50 × 10^−5^	1.50 × 10^−4^; 1.35 × 10^−4^; 1.20 × 10^−4^; 1.05 × 10^−4^; 9.00 × 10^−5^; 7.50 × 10^−5^; 6.00 × 10^−5^; 4.50 × 10^−5^; 3.00 × 10^−5^; 1.50 × 10^−5^; 7.50 × 10^−6^; 1,50 × 10^−6^; 7.50 × 10^−7^

QAC 1—benzyl-C12-18-alkydimethyl ammonium chlorides; QAC 2—benzyl-C12-16 alkyldimethyl chlorides; QAC 3—didecyldimethylammonium chloride, benzyl-C12-16-alkyldimethyl chlorides; OA 1—hydrogen peroxide, silver nitrate; OA 2—perlactic acid; OA 3—peracetic acid, hydrogen peroxide bis (sulphate) bis (peroxymonosulfate); OA 4—pentapotassium, benzenesulfonic acid, C10-13 alkyl derivatives, sodium salts, malic acid, sulfamic acid; ChC 1—chlorine dioxide; ChC 2—hypochlorous acid calcium salt; IC 1—iodine, IC 2—iodine; NANO 1—nanocopper; NANO 2—nanosilver.

**Table 3 microorganisms-07-00127-t003:** Minimal bactericidal concentration of tested disinfectant depending of water type (g/mL).

Group of Disinfectants	Disinfectant	Water Type	Minimal Bactericidal Concentration of Disinfectant (g/mL)					
			Strain					
			LMO-ATCC	LMO-W	LMO-M	LMO-N	LMO-R	LMO-K
**Quaternary ammonium compounds**	**QAC 1**	Nonozonated	1.00 × 10^−4 a^	1.00 × 10^−4 a^	4.00 × 10^−5 b^	4.00 × 10^−5 b^	4.00 × 10^−5 b^	1.00 × 10^−4 a^
		Ozonated	4.00 × 10^−5 b^	2.00 × 10^−5 b^	1.00 × 10^−5 b^	1.00 × 10^−5 b^	1.00 × 10^−5 b^	4.00 × 10^−5 b^
	QAC 2	Nonozonated	1.28 × 10^−1 c^	1.28 × 10^−1 c^	1.28 × 10^−1 c^	1.28 × 10^−1 c^	1.28 × 10^−1 c^	1.28 × 10^−1 c^
		Ozonated	1.28 × 10^−2 d^	1.28 × 10^−2 d^	1.28 × 10^−2 d^	1.28 × 10^−2 d^	1.28 × 10^−2 d^	1.28 × 10^−2 d^
	QAC 3	Nonozonated	1.49 × 10^−1 c^	1.49 × 10^−1 c^	1.49 × 10^−1 c^	1.49 × 10^−1 c^	1.49 × 10^−1 c^	1.49 × 10^−1 c^
		Ozonated	1.49 × 10^−2 d^	2.97 × 10^−2 d^	1.49 × 10^−2 d^	1.49 × 10^−2 d^	1.49 × 10^−2 d^	1.49 × 10^−2 d^
**Oxidizing agents**	OA 1	Nonozonated	1.20 × 10^1 e^	1.20 × 10^1 e^	1.20 × 10^1 e^	1.20 × 10^1 e^	1.20 × 10^1 e^	1.20 × 10^1 e^
		Ozonated	1.01 × 10^1 e^	1.01 × 10^1 e^	1.01 × 10^1 e^	1.01 × 10^1 e^	1.01 × 10^1 e^	1.01 × 10^1 e^
	OA 2	Nonozonated	4.90 × 10^0 k^	Ineffective	Ineffective	4.90 × 10^0 k^	Ineffective	4.90 × 10^0 k^
		Ozonated	2.70 × 10^0 f^	2.79 × 10^0 f^	3.38 × 10^0 f,k^	2.70 × 10^0 f^	3.68 × 10^0 f,k^	2.70 × 10^0 f^
	OA 3	Nonozonated	1.42 × 10^−3 g^	1.42 × 10^−3 g^	1.85 × 10^−3 g^	1.85 × 10^−3 g^	3.69 × 10^−3 g^	1.42 × 10^−3 g^
		Ozonated	3.08 × 10^−4 a^	3.08 × 10^−4 a^	4.31 × 10^−4 a^	3.08 × 10^−4 a^	5.54 × 10^−4 a^	3.08 × 10^−4 a^
	OA 4	Nonozonated	4.20 × 10^−3 h^	4.20 × 10^−3 h^	4.20 × 10^−3 h^	4.20 × 10^−3 h^	4.20 × 10^−3 h^	4.20 × 10^−3 h^
		Ozonated	1.60 × 10^−3 g^	1.60 × 10^−3 g^	1.60 × 10^−3 g^	1.60 × 10^−3 g^	1.60 × 10^−3 g^	1.60 × 10^−3 g^
**Chlorine compounds**	ChC 1	Nonozonated	2.00 × 10^−7 i^	4.00 × 10^−7 i^	7.00 × 10^−7 i^	4.00 × 10^−7 i^	6.00 × 10^−7 i^	4.00 × 10^−7 i^
		Ozonated	5.00 × 10^−8 j^	1.00 × 10^−7 i^	3.00 × 10^−7 i^	1.00 × 10^−7 i^	2.00 × 10^−7 i^	1.00 × 10^−7 i^
	ChC 2	Nonozonated	2.40 × 10^−1 c^	3.20 × 10^−1 c^	3.60 × 10^−1 c^	3.20 × 10^−1 c^	2.80 × 10^−1 c^	4.0 × 10^−1 c^
		Ozonated	2.00 × 10^−1 c^	2.00 × 10^−1 c^	2.00 × 10^−1 c^	2.00 × 10^−1 c^	2.00 × 10^−1 c^	2.80 × 10^−1 c^
**Iodine compounds**	IC 1	Nonozonated	2.15 × 10^0 f^	2.15 × 10^0 f^	2.15 × 10^0 f^	2.15 × 10^0 f^	2.15 × 10^0 f^	2.15 × 10^0 f^
		Ozonated	1.05 × 10^0 l^	1.29 × 10^0 l^	1.29 × 10^0 l^	1.29 × 10^0 l^	1.29 × 10^0 l^	1.05 × 10^0 l^
	IC 2	Nonozonated	3.63 × 10^0 f,k^	3.63 × 10^0 f,k^	3.86 × 10^0 f,k^	3.63 × 10^0 f,k^	3.63 × 10^0 f,k^	3.86 × 10^0 f,k^
		Ozonated	1.64 × 10^0 l^	1.64 × 10^0 l^	1.76 × 10^0 l^	1.64 × 10^0 l^	1.69 × 10^0 l^	1.76 × 10^0 l^
**Nanoparticles**	NANO 1	Nonozonated	1.37 × 10^−4 a^	1.37 × 10^−4 a^	1.41 × 10^−4 a^	1.37 × 10^−4 a^	1.47 × 10^−4 a^	1.37 × 10^−4 a^
		Ozonated	1.08 × 10^−4 a^	1.14 × 10^−4 a^	1.14 × 10^−4 a^	1.08 × 10^−4 a^	1.19 × 10^−4 a^	1.08 × 10^−4 a^
	NANO 2	Nonozonated	Ineffective	Ineffective	Ineffective	Ineffective	Ineffective	Ineffective
		Ozonated	1.20 × 10^−4 a^	1.20 × 10^−4 a^	1.20 × 10^−4 a^	1.20 × 10^−4 a^	1.20 × 10^−4 a^	1.20 × 10^−4 a^

QAC 1—benzyl-C12-18-alkydimethyl ammonium chlorides; QAC 2—benzyl-C12-16 alkyldimethyl chlorides; QAC 3—didecyldimethylammonium chloride, benzyl-C12-16-alkyldimethyl chlorides; OA 1—hydrogen peroxide, silver nitrate; OA 2—perlactic acid; OA 3—peracetic acid, hydrogen peroxide bis (sulphate) bis (peroxymonosulfate); OA 4—pentapotassium, benzenesulfonic acid, C10-13 alkyl derivatives, sodium salts, malic acid, sulfamic acid; ChC 1—chlorine dioxide; ChC 2—hypochlorous acid calcium salt; IC 1—iodine, IC 2—iodine; NANO 1—nanocopper; NANO 2—nanosilver; LMO-ATCC—*L. monocytogenes* ATCC 19111, LMO-W—strain isolated from vegetables, LMO-M—strain isolated from meat, LMO-N—strain isolated from dairy products, LMO-R—strain isolated from fish, LMO-K—clinical strain, a–l—values marked with different letters differ statistically significantly (*p* ≤ 0.05).

**Table 4 microorganisms-07-00127-t004:** Logarithmic decreases in bacteria number after using the disinfectant concentration directly preceding MBC (preMBC).

Group of Disinfectants	Disinfectant	Water Type	Reduction in Bacteria Number (log CFU)					
			Strain					
			LMO-ATCC	LMO-W	LMO-M	LMO-N	LMO-R	LMO-K
**Quaternary ammonium compounds**	QAC 1	**Initial no.**	**8.81**	**8.69**	**8.71**	**8.76**	**8.66**	**8.73**
		Nonozonated	7.23 ^a^	6.96 ^a^	6.84 ^a^	6.92 ^a^	6.76 ^a^	7.07 ^a^
		Ozonated	8.28 ^b^	8.00 ^b^	7.88 ^b^	7.97 ^b^	7.80 ^b^	8.12 ^b^
	QAC 2	**Initial no.**	**8.81**	**8.69**	**8.71**	**8.76**	**8.66**	**8.73**
		Nonozonated	7.05 ^a^	6.78 ^a^	6.62 ^a^	6.75 ^a^	6.50 ^a^	6.90 ^a^
		Ozonated	7.93 ^b^	7.65 ^b^	7.62 ^b^	7.49 ^a^	7.36 ^b^	7.77 ^b^
	QAC 3	**Initial no.**	**8.81**	**8.69**	**8.71**	**8.76**	**8.66**	**8.73**
		Nonozonated	6.70 ^a^	6.43 ^a^	6.27 ^a^	6.40 ^a^	6.15 ^a^	6.55 ^a^
		Ozonated	7.49 ^a^	7.22 ^a^	7.06 ^a^	7.19 ^a^	6.93 ^a^	7.34 ^a^
**Oxidizing agents**	OA 1	**Initial no.**	**8.81**	**8.69**	**8.71**	**8.76**	**8.66**	**8.73**
		Nonozonated	6.52 ^a^	6.26 ^a^	6.10 ^a^	6.22 ^a^	5.98 ^a^	6.37 ^a^
		Ozonated	7.23 ^b^	6.96 ^a^	6.79 ^a^	6.93 ^a^	6.67 ^a^	7.07 ^a^
	OA 2	**Initial no.**	**8.81**	**8.69**	**8.71**	**8.76**	**8.66**	**8.73**
		Nonozonated	2.64 ^a^	2.43 ^a^	2.37 ^a^	2.17 ^a^	2.26 ^a^	2.53 ^a^
		Ozonated	5.11 ^b^	4.87 ^b^	4.59 ^b^	4.82 ^b^	4.70 ^b^	4.98 ^b^
	OA 3	**Initial no.**	**8.81**	**8.69**	**8.71**	**8.76**	**8.66**	**8.73**
		Nonozonated	6.61 ^a^	6.35 ^a^	6.18 ^a^	6.31 ^a^	6.06 ^a^	6.46 ^a^
		Ozonated	7.40 ^a^	7.13 ^a^	6.97 ^a^	7.08 ^a^	6.80 ^a^	7.25 ^a^
	OA 4	**Initial no.**	**8.81**	**8.69**	**8.71**	**8.76**	**8.66**	**8.73**
		Nonozonated	6.43 ^a^	6.18 ^a^	6.02 ^a^	6.12 ^a^	5.83 ^a^	6.29 ^a^
		Ozonated	7.35 ^b^	7.04 ^b^	6.90 ^b^	7.00 ^b^	6.75 ^b^	7.10 ^b^
**Chlorine compounds**	ChC 1	**Initial no.**	**8.81**	**8.69**	**8.71**	**8.76**	**8.66**	**8.73**
		Nonozonated	7.67 ^a^	7.39 ^a^	7.23 ^a^	7.33 ^a^	7.06 ^a^	7.52 ^a^
		Ozonated	8.55 ^b^	8.30 ^b^	8.24 ^b^	8.10 ^a^	7.91 ^b^	8.33 ^b^
	ChC 2	**Initial no.**	**8.81**	**8.69**	**8.71**	**8.76**	**8.66**	**8.73**
		Nonozonated	7.31 ^a^	6.95 ^a^	6.71 ^a^	6.93 ^a^	6.58 ^a^	7.08 ^a^
		Ozonated	7.85 ^a^	7.56 ^a^	7.30 ^a^	7.55 ^a^	7.41 ^b^	7.65 ^a^
**Iodine compounds**	IC 1	**Initial no.**	**8.81**	**8.69**	**8.71**	**8.76**	**8.66**	**8.73**
		Nonozonated	6.48 ^a^	6.22 ^a^	6.05 ^a^	6.16 ^a^	5.93 ^a^	6.37 ^a^
		Ozonated	7.54 ^b^	7.26 ^b^	7.17 ^b^	7.31 ^b^	7.03 ^b^	7.38 ^b^
	IC 2	**Initial no.**	**8.81**	**8.69**	**8.71**	**8.76**	**8.66**	**8.73**
		Nonozonated	6.49 ^a^	6.21 ^a^	6.03 ^a^	6.15 ^a^	5.90 ^a^	6.33 ^a^
		Ozonated	7.63 ^b^	7.34 ^b^	7.16 ^b^	7.30 ^b^	7.04 ^b^	7.47 ^b^
**Nanoparticles**	NANO 1	**Initial no.**	**8.81**	**8.69**	**8.71**	**8.76**	**8.66**	**8.73**
		Nonozonated	6.17 ^a^	5.91 ^a^	5.75 ^a^	5.87 ^a^	5.69 ^a^	6.03 ^a^
		Ozonated	7.05 ^b^	6.79 ^b^	6.62 ^b^	6.75 ^b^	6.50 ^b^	6.90 ^b^
	NANO 2	**Initial no.**	**8.81**	**8.69**	**8.71**	**8.76**	**8.66**	**8.73**
		Nonozonated	1.76 ^a^	1.57 ^a^	1.30 ^a^	1.49 ^a^	1.37 ^a^	1.66 ^a^
		Ozonated	6.03 ^b^	5.75 ^b^	5.87 ^b^	5.91 ^b^	5.63 ^b^	6.17 ^b^

QAC 1—benzyl-C12-18-alkydimethyl ammonium chlorides; QAC 2—benzyl-C12-16 alkyldimethyl chlorides; QAC 3—didecyldimethylammonium chloride, benzyl-C12-16-alkyldimethyl chlorides; OA 1—hydrogen peroxide, silver nitrate; OA 2—perlactic acid; OA 3—peracetic acid, hydrogen peroxide bis (sulphate) bis (peroxymonosulfate); OA 4—pentapotassium, benzenesulfonic acid, C10-13 alkyl derivatives, sodium salts, malic acid, sulfamic acid; ChC 1—chlorine dioxide; ChC 2—hypochlorous acid calcium salt; IC 1—iodine, IC 2—iodine; NANO 1—nanocopper; NANO 2—nanosilver; LMO-ATCC—*L. monocytogenes* ATCC 19111, LMO-W—strain isolated from vegetables, LMO-M—strain isolated from meat, LMO-N—strain isolated from dairy products, LMO-R—strain isolated from fish, LMO-K—clinical strain, a,b—values marked with different letters differ statistically significantly (*p* ≤ 0.05) (Tested separately for each disinfectant and each strain depending on water type—ozonated/non-ozonated).

**Table 5 microorganisms-07-00127-t005:** Efficiency coefficient values for tested strains and disinfectants.

Disinfectant	Efficiency Coefficient (A)					
	Strain					
	LMO-ATCC	LMO-W	LMO-M	LMO-N	LMO-R	LMO-K
QAC 1	0.40 ^a^	0.20 ^b^	0.25 ^b,g^	0.25 ^b,g^	0.25 ^b,g^	0.40 ^a^
QAC 2	0.10 ^c,f^	0.10 ^c,f^	0.10 ^c,f^	0.10 ^c,f^	0.10 ^c,f^	0.10 ^c,f^
QAC 3	0.10 ^c,f^	0.20 ^b^	0.10 ^c,f^	0.10 ^c,f^	0.10 ^c,f^	0.10 ^c,f^
OA 1	0.84 ^d^	0.84 ^d^	0.84 ^d^	0.84 ^d^	0.84 ^d^	0.84 ^d^
OA 2	0.55 ^e,i^	Ineffective	Ineffective	0.55 ^e,i^	Ineffective	0.55 ^e,i^
OA 3	0.22 ^b^	0.22 ^b^	0.23 ^b^	0.17 ^b,f^	0.15 ^b,f^	0.22 ^b^
OA 4	0.38 ^a^	0.38 ^a^	0.38 ^a^	0.38 ^a^	0.38 ^a^	0.38 ^a^
ChC 1	0.25 ^b,g^	0.25 ^b,g^	0.43 ^a^	0.25 ^b,g^	0.33 ^a,g^	0.25 ^b,g^
ChC 2	0.83 ^d^	0.63 ^e,h^	0.56 ^e^	0.63 ^e,h^	0.71 ^h,j^	0.70 ^h,j^
IC 1	0.49 ^a,i^	0.60 ^e,h^	0.60 ^e,h^	0.60 ^e,h^	0.60 ^e,h^	0.49 ^a,i^
IC 2	0.45 ^a,i^	0.45 ^a,i^	0.45 ^a,i^	0.45 ^a,i^	0.45 ^a,i^	0.45 ^a,i^
NANO 1	0.70 ^h,j^	0.84 ^d^	0.81 ^d^	0.79 ^d,j^	0.81 ^d^	0.79 ^d,j^
NANO 2	Ineffective	Ineffective	Ineffective	Ineffective	Ineffective	Ineffective

QAC 1—benzyl-C12-18-alkydimethyl ammonium chlorides; QAC 2—benzyl-C12-16 alkyldimethyl chlorides; QAC 3—didecyldimethylammonium chloride, benzyl-C12-16-alkyldimethyl chlorides; OA 1—hydrogen peroxide, silver nitrate; OA 2—perlactic acid; OA 3—peracetic acid, hydrogen peroxide bis (sulphate) bis (peroxymonosulfate); OA 4—pentapotassium, benzenesulfonic acid, C10-13 alkyl derivatives, sodium salts, malic acid, sulfamic acid; ChC 1—chlorine dioxide; ChC 2—hypochlorous acid calcium salt; IC 1—iodine, IC 2—iodine; NANO 1—nanocopper; NANO 2—nanosilver; LMO-ATCC—*L. monocytogenes* ATCC 19111, LMO-W—strain isolated from vegetables, LMO-M—strain isolated from meat, LMO-N—strain isolated from dairy products, LMO-R—strain isolated from fish, LMO-K—clinical strain, a–j—values marked with different letters differ statistically significantly (*p* ≤ 0.05).

**Table 6 microorganisms-07-00127-t006:** Stability assessment of stored solutions of QAC 2, OA 3, and ChC 1 after 0. 12 and 24 h.

Disinfectant	Water Type	Storage Time (h)	Minimal Bactericidal Concentration of Disinfectant [g/cm^3^]					
			Strain					
			LMO-ATCC	LMO-W	LMO-M	LMO-N	LMO-R	LMO-K
**QAC 2 (working solution: 2.55 × 10^0^ g/mL)**	**Nonozonated**	0	1.28 × 10^−4 a^	1.28 × 10^−4 a^	1.28 × 10^−4 a^	1.28 × 10^−4 a^	1.28 × 10^−4 a^	1.28 × 10^−4 a^
		12	1.28 × 10^−4 a^	1.28 × 10^−4 a^	1.28 × 10^−4 a^	1.28 × 10^−4 a^	1.28 × 10^−4 a^	1.28 × 10^−4 a^
		24	1.53 × 10^−4 a^	1.53 × 10^−4 a^	1.53 × 10^−4 a^	1.53 × 10^−4 a^	1.53 × 10^−4 a^	1.53 × 10^−4 a^
	Ozonated	0	1.28 × 10^−5 b^	1.28 × 10^−5 b^	1.28 × 10^−5 b^	1.28 × 10^−5 b^	1.28 × 10^−5 b^	1.28 × 10^−5 b^
		12	1.02 × 10^−4 a^	1.28 × 10^−5 b^	1.28 × 10^−5 b^	1.28 × 10^−5 b^	1.28 × 10^−5 b^	1.02 × 10^−4 a^
		24	1.53 × 10^−4 a^	1.53 × 10^−4 a^	1.53 × 10^−4 a^	1.53 × 10^−4 a^	1.79 × 10^−4 a^	1.53 × 10^−4 a^
**OA 3 (working solution: 6.15 × 10^−3^ g/mL)**	Nonozonated	0	1.42 × 10^−3 c^	1.42 × 10^−3 c^	1.42 × 10^−3 c^	1.80 × 10^−3 c^	3.81 × 10^−3 d,f^	1.42 × 10^−3 c^
		12	1.66 × 10^−3 c^	1.72 × 10^−3 c^	2.40 × 10^−3 c,d^	2.20 × 10^−3 c,d^	4.30 × 10^−3 e,f^	1.60 × 10^−3 c^
		24	2.40 × 10^−3 c,d^	2.58 × 10^−3 c,d^	3.40 × 10^−3 d^	3.20 × 10^−3 d^	5.23 × 10^−3 e^	2.50 × 10^−3 c,d^
	Ozonated	0	3.10 × 10^−4 g^	3.10 × 10^−4 g^	4.30 × 10^−4 g^	3.10 × 10^−4 g^	6.15 × 10^−4 h^	2.46 × 10^−4 g^
		12	1.60 × 10^−3 c^	1.66 × 10^−3 c^	2.40 × 10^−3 c,d^	2.15 × 10^−3 c,d^	4.12 × 10^−3 e,f^	1.60 × 10^−3 c^
		24	2.40 × 10^−3 c,d^	2.65 × 10^−3 c,d^	3.40 × 10^−3 d^	3.9 × 10^−3 d,f^	5.10 × 10^−3 e^	2.50 × 10^−3 c,d^
**ChC 1 (working solution: 1.00 × 10^−5^ g/mL)**	Nonozonated	0	2.00 × 10^−7 i^	4.00 × 10^−7 i,k^	7.00 × 10^−7 j^	4.00 × 10^−7 i,k^	6.00 × 10^−7 j,k^	4.00 × 10^−7 i,k^
		12	3.00 × 10^−7 i^	6.00 × 10^−7 j,k^	8.00 × 10^−7 j^	5.00 × 10^−7 i,k^	8.00 × 10^−7 j^	5.00 × 10^−7 i,k^
		24	5.00 × 10^−7 i,k^	8.00 × 10^−7 j^	1.00 × 10^−6 l^	7.00 × 10^−7 j^	1.10 × 10^−6 l^	7.00 × 10^−7 j^
	Ozonated	0	5.00 × 10^−8 m^	1.00 × 10^−7 i^	3.00 × 10^−7 i^	1.00 × 10^−7 i^	2.00 × 10^−7 i^	1.00 × 10^−7 i^
		12	2.00 × 10^−7 i^	5.00 × 10^−7 i,k^	8.00 × 10^−7 j^	5.00 × 10^−7 i,k^	8.00 × 10^−7 j^	4.00 × 10^−7 i,k^
		24	5.00 × 10^−7 i,k^	8.00 × 10^−7 j^	1.00 × 10^−6 l^	8.00 × 10^−7 j^	1.10 × 10^−6 l^	7.00 × 10^−7 j^

QAC 2—benzyl-C12-16 alkyldimethyl chlorides; OA 3—peracetic acid, hydrogen peroxide bis (sulphate) bis (peroxymonosulfate); ChC 1—chlorine dioxide; LMO-ATCC—*L. monocytogenes* ATCC 19111, LMO-W—strain isolated from vegetables, LMO-M—strain isolated from meat, LMO-N—strain isolated from dairy products, LMO-R—strain isolated from fish, LMO-K—clinical strain, a–m—values marked with different letters differ statistically significantly (*p* ≤ 0.05).

**Table 7 microorganisms-07-00127-t007:** Efficiency coefficient values for tested strains and disinfectants after storage of solutions.

Disinfectant	Storage Time (h)	Efficiency Coefficient (A)					
		Strain					
		LMO-ATCC	LMO-W	LMO-M	LMO-N	LMO-R	LMO-K
QAC 2 (working solution: 2.55 × 10^0^ g/mL)	0 12 24	0.10 ^a^ 0.80 ^b^ 1.00 ^b,c^	0.10 ^a^ 0.10 ^a^ 1.00 ^b,c^	0.10 ^a^ 0.10 ^a^ 1.00 ^b,c^	0.10 ^a^ 0.10 ^a^ 1.00 ^b,c^	0.10 ^a^ 0.10 ^a^ 1.17 ^c^	0.10 ^a^ 0.80 ^b^ 1.00 ^b,c^
OA 3 (working solution: 6.15 × 10^−3^ g/mL)	0 12 24	0.22 ^a^ 0.96 ^b^ 1.00 ^b,c^	0.22 ^a^ 0.96 ^b^ 1.02 ^b,c^	0.23 ^a^ 1.00 ^b,c^ 1.00 ^b,c^	0.17 ^a^ 0.97 ^b^ 1.06 ^b,c^	0.16 ^a^ 0.96 ^b^ 0.98 ^b,c^	0.17 ^a^ 1.00 ^b,c^ 1.00 ^b,c^
ChC 1 (working solution: 1.00 × 10^−5^ g/mL)	0 12 24	0.25 ^a,e^ 0.67 ^f^ 1.00 ^b,c^	0.25 ^a^ 0.83 ^b^ 1.00 ^b,c^	0.43 ^d^ 1.00 ^b,c^ 1.00 ^b,c^	0.25 ^a,e^ 1.00 ^b,c^ 1.14 ^b,c^	0.33 ^d,e^ 1.00 ^b,c^ 1.00 ^b,c^	0.25 ^a,e^ 0.80 ^b^ 1.00 ^b,c^

QAC 2—benzyl-C12-16 alkyldimethyl chlorides; OA 3—peracetic acid, hydrogen peroxide bis (sulphate) bis (peroxymonosulfate); ChC 1—chlorine dioxide; LMO-ATCC—*L. monocytogenes* ATCC 19111, LMO-W—strain isolated from vegetables, LMO-M—strain isolated from meat, LMO-N—strain isolated from dairy products, LMO-R—strain isolated from fish, LMO-K—clinical strain, a–f—values marked with different letters differ statistically significantly (*p* ≤ 0.05).

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
