# Peer review of "Biocidal Effectiveness of Selected Disinfectants Solutions Based on Water and Ozonated Water against Listeria monocytogenes Strains"

_microorganisms, 2019, doi:10.3390/microorganisms7050127_

Round 1

Reviewer 1 Report

Finding effective biocidal for Listeria monocytogenes is always important. This paper has importance, but I have the following concerns which need to be addressed... 

- The abstract is not clearly focusing on the significance of the study,  needs to rewrite.

- line 19 Listeria needs of rewrite into the full form. Write all the names of six L. monocytogenes into the abstract. In abstract QAC, ChC, etc need to write the full form together inside the bracket.

- In materials and methods, the bacterial strains are used in this study is there any similarities among the other outbreak strains of L. monocytogenes

- line 90 check the spelling of 'reated'

- As ozone has half lifetime in water how the pH was maintained and was there any changes with temperature increase?

- in table 1 how these working solution concentrations were selected?

- line 100 and101 how these concentrations (200%, 180%, 160% and 120%) were selected?

- in table 2 initial and final concentrations how and why these concentrations were selected these information needs to be more clarified into the paper.

Preparation of bacterial suspensions needs to write step by step procedure description. The bacterial suspension is the mixture of all the isolates or it has done separately is not clear. For Listeria growth why Columbia agar media used instead of tryptic soy agar media and PALCAM media?

- Line 112 do not start with numerical (100) and Eppendorf tubes size and company name information are missing.

-Line 115 how the bacterial CFU was determined?

-line 125, write the full form of CAB

-line 136, do not start with numerical.

- line 160, what does that 1.1 mean?

- Fig 1 and 2 legends need to be more elaborated and described to make it clearly understandable.

- For this experiment why distilled water didn't use to compare?

- Another one or two figures should be added where it shows the log reduction of Listeria with different ozonated disinfectants. 

Author Response

Poland, April 19, 2019

Dear Reviewer,

We are very grateful for your time and valuable remarks, which will certainly help to improve the quality of the manuscript.

The changes introduced in the text of the manuscript have been marked with yellow highlights.

Detailed response to the review is presented below:

Finding effective biocidal for Listeria monocytogenes is always important. This paper has importance, but I have the following concerns which need to be addressed.

- The abstract is not clearly focusing on the significance of the study,  needs to rewrite.

Abstract has been rewrite. Information about significance and novelity of research has been added (lines 37-41)

- line 19 Listeria needs of rewrite into the full form. Write all the names of six L. monocytogenes into the abstract. In abstract QAC, ChC, etc need to write the full form together inside the bracket.

These changes has been made (lines 20-22, 22-24 and 29-33)

- In materials and methods, the bacterial strains are used in this study is there any similarities among the other outbreak strains of L. monocytogenes

 The strains used in this study had an examined level of genetic similarity with the PFGE method as part of our previous studies. These strains were not genetically identical. They came from routine research and were not associated with any epidemic.

- line 90 check the spelling of 'reated'

It should be „treated” – it has been corrected (line 101)

- As ozone has half lifetime in water how the pH was maintained and was there any changes with temperature increase?

In our research, we did not allow temperature changes, so we do not know what the impact of such changes would be. Unfortunately, pH measurement was not performed or we did not use any buffers.

- in table 1 how these working solution concentrations were selected?

The initial working solution concentrations presented in table 1 are in accordance with the manufacturer's instructions for particular disinfecntant – this information has been added (lines 106-107)

- line 100 and 101 how these concentrations (200%, 180%, 160% and 120%) were selected?

We have described it in manuscript (lines 110-114)

- in table 2 initial and final concentrations how and why these concentrations were selected these information needs to be more clarified into the paper.

We have added this information (lines 113-114)

- Preparation of bacterial suspensions needs to write step by step procedure description. The bacterial suspension is the mixture of all the isolates or it has done separately is not clear. For Listeria growth why Columbia agar media used instead of tryptic soy agar media and PALCAM media?

Preparation of bacterial suspensions has been described more detailed (lines 127-135). Our previous research showed that using Columbia agar with 5% sheep blood allows more intensive and faster growth of L. monocytogenes than with the use of TSA. This is especially important when trying to recover bacterial cells after environmental stress. In turn, the use of strains grown on selective media, such as PALCAM, for the efficacy of antimicrobial activity is not recommended.

- Line 112 do not start with numerical (100) and Eppendorf tubes size and company name information are missing.

It has been corrected and information have been added (lines 137-138).

-Line 115 how the bacterial CFU was determined?

It has been described (lines 140-144)

-line 125, write the full form of CAB

It has been corrected (line 143)

-line 136, do not start with numerical.

It has been corrected (line 169)

- line 160, what does that 1.1 mean?

It should be 2.7 (line 196)

- Fig 1 and 2 legends need to be more elaborated and described to make it clearly understandable.

It has been corrected (lines 229-233 and 262-264)

- For this experiment why distilled water didn't use to compare?

We have demonstrated in point 3.1. that water was ineffective against tested strains, so in point 3.2. we have only compared effectiveness of disinfectants solutions prepared based on ozonated and non-ozonated water.

- Another one or two figures should be added where it shows the log reduction of Listeria with different ozonated disinfectants. 

Unfortunately, we will not be able to prepare this type of figures. In our studies, we decided to treat as the MBC value concentration at which no growth of even a single colony of the tested strains was found. For this reason, after disinfection and neutralization, we sowed the material on an agar without disinfectants. On this medium, we made a linear seeding and incubated cultures. We evaluated the result in the zero-one form - the presence of growth / lack of growth. We did not count colonies.

Reviewer 2 Report

I find the study has high applied implications. However, there are few language issues that need to be amended/fixed. The statistical analysis and the experimental planning (number of experiments, measurements and how the results were handled) need to be described in detail. I made significant number of comments and suggested changes on the PDF file to help your revision in addition to the comments below.

L26-30 please introduce the compounds first (not the abbreviation).

It is not clear what the numbers associated with the abbreviations mean (OC3)!! What this 3 refers to?

Please use g/mL for concentrations as its more clear than g/cm3

L51 and L59  the use of different units for Ozone treatment is quite difficult to follow. Please use consistent units to help the reader to understand, or if this is not possible, then some background about how the units logic need to be provided.

Author Response

Poland, April 19, 2019

Dear Reviewer,

We are very grateful for your time and valuable remarks, which will certainly help to improve the quality of the manuscript.

The changes introduced in the text of the manuscript have been marked with yellow highlights.

Detailed response to the review is presented below:

I find the study has high applied implications. However, there are few language issues that need to be amended/fixed. The statistical analysis and the experimental planning (number of experiments, measurements and how the results were handled) need to be described in detail. I made significant number of comments and suggested changes on the PDF file to help your revision in addition to the comments below.

L26-30 please introduce the compounds first (not the abbreviation).

It has been corrected (lines 22-24)

It is not clear what the numbers associated with the abbreviations mean (OC3)!! What this 3 refers to?

It has been explained in text (lines 29-33)

Please use g/mL for concentrations as its more clear than g/cm3

It has been changed in whole manuscript

L51 and L59  the use of different units for Ozone treatment is quite difficult to follow. Please use consistent units to help the reader to understand, or if this is not possible, then some background about how the units logic need to be provided.

These data comes from the work of other authors, so we think that we have no right to make changes to their data. If necessary, we can convert the units.

Comments from PDF file:

Line 70 – please explain what is hard water means here (Tape water or specially prepared water) ?

This specially prepared water. It has been explained in text - lines 93-94

Line 81 – spelling

It has been corrected - line 90

Line 84 – what this code refer to

It has been explained - lines 93-94

Line 85 – please provide full information for the source of this instrument (company, city, country)

It has been added - line 95

Line 88 – please provide full names for any abbreviations at first mentioning

It has been corrected - lines 97-98

Line 88 – language

It is ok

Line 89 – please provide full details for suppliers

It has been provided - line 99

Line 90 – language

It has been corrected - line 101

Line 95 – spelling

It has been corrected - line 105

Line 96 -.

It has been corrected - line 106

Line 99 – please state the source of these recommendations and justify the use of the concentrations in table 1

It has been explained in text - lines 110-114

Table 2 – this state very high solubization (almost 10 g in 1 ml) is this correct? It moght be good to have a solubilization level added for these compounds

This is correct. In the product characteristics sheets, the manufacturer has posted, for all these disinfectants, a record: completely miscible / soluble in water

Line 112

It has been corrected - line 137

Line 113 – at what conncentrations ? and how it was adjusted for specific concentrations ?

It has been explained in text - lines 138-140

Line 116- please be consistent describing the water (hard, plain, … etc) –

We have changed in whole manuscript to „sterile hard water”

Lines 119-120 – please provide full information

It has been prowided - lines 149-152

Line 120 – distilled, deionized, hard, tape … ?

We have changed in whole manuscript to „sterile hard water”

Lines 122-123 – where this ethanol used/came from ?

The composition of the neutralizer is given in the Polish Standard and this must be used in disinfection studies. We have described the inhibitory actions of all components of the neutralizer. For this reason, we also listed substances that affect other disinfectants than those included in our research, such as ethanol.

Line 135 – wrong word. Evaluation not Evolution

It has been corrected - line 168

Line 136

It has been corrected - line 169

Line 144 – what is this ?

It has been explained - Line 178

Line 170-171 – this is not sufficient description of the statistical analysis. Please state what model was used ? How the analysis was done (what test was carried out) ? what were the dependent and independent variables in the model… -

The statistical analysis description has been improved - Lines 206-216

Line 180 – Please include the variation

It has been added – lines 221-224

Line 184 – all abbreviations need to be explained in the figure caption. There are no indication of statistical analysis. There is no non-treated control. Where 2 h bars ?

All suggestions have been taken into account – lines 227-233

However, we could not show the results for a negative control that would be sterile hard water. The figure 1 does not show changes in the number of bacteria. This figure shows the changes of MBC of ozone in ozonated water over time. We are not able to present an effective water concentration as a negative control.

Line 185

It has been corrected - line 234

Line 200 – tables and figures should stand alone. Therfore, any abbreviations need to be explained as footnotes

All abbreviations have been explained - lines 249-253

Line 203 – its not easy to differentiate between negative or positive effects using this way

It has been corrected - lines 255- 258

Fig. 2 – no error bars

Fig.2. has been corrected

Line 211

It has been corrected - line 264

Line 225 – stability

It has been replaced – line 287

Round 2

Reviewer 1 Report

This paper has importance to the readers and also has a great potentiality to be published, but I have following concerns-

As Listeria monocytogenes can survive in only water more than 4 weeks without any supplement and also there are possibilities to have background microflora in sources. So before recommending any product to the food industry to control any microorganism, it is really important to know the exact reduction of the microorganisms. So before publishing this paper my suggestion to conduct experiments to show the log reduction of Listeria monocytogenes with all of the disinfectants, using selective and non-selective media. This log reduction along with MBC values can explain the result statement very effectively. Because even 0.01% more chance of surviving of the pathogens can cause a great problem in the food processing industry. So before giving any recommendation it is important to have those experiments information.

Also,

- Recovering L. monocytogenes from synergism effect of elevated temperature, pressure and  antomicrobials on TSA is very good. So this media can be used as an alternative.

- Line 19 write the full form listeria.

Author Response

 Poland, May 1, 2019

Dear Reviewer,

We are very grateful for your time and valuable remarks, which will certainly help us to improve the quality of the manuscript.

The changes introduced in the text of the manuscript have been marked with yellow highlights.

Detailed response to the review is presented below:

Reviewer 1

This paper has importance to the readers and also has a great potentiality to be published, but I have following concerns:

As Listeria monocytogenes can survive in only water more than 4 weeks without any supplement and also there are possibilities to have background microflora in sources. So before recommending any product to the food industry to control any microorganism, it is really important to know the exact reduction of the microorganisms. So before publishing this paper my suggestion to conduct experiments to show the log reduction of Listeria monocytogenes with all of the disinfectants, using selective and non-selective media. This log reduction along with MBC values can explain the result statement very effectively. Because even 0.01% more chance of surviving of the pathogens can cause a great problem in the food processing industry. So before giving any recommendation it is important to have those experiments information.

We agree with the Reviewer that determining the logarithmic decreases in the number of L. monocytogenes is very valuable information. However, in all cases, the concentration of disinfectants equal MBC value cause decrease in bacteria number below the detection limit. For this reason, we decided to present logarithmic decreases in the number of L. monocytogenes determined for the concentration directly preceding the MBC - we called them preMBC. Detailed information about the calculation of inheritance is presented in the section of the methodology (lines 199-214), and the results in table 4 in the text in lines 291-298.

Also,

- Recovering L. monocytogenes from synergism effect of elevated temperature, pressure and  antomicrobials on TSA is very good. So this media can be used as an alternative.

We agree with the Reviewer's comments. TSA is very good agar medium, however, we conducted our research using Columbia Agar with sheep blood, which is also acceptable.

- Line 19 write the full form listeria.

It has been corrected.

Best regards,

Krzysztof Skowron

Reviewer 2 Report

The statistical analysis is not clear. As per earlier comments, please provide the model used for MANOVA stating the dependent and independent variables. Was there any repeat measurements in the design?

The Polish Standard PN-EN 1276:2010 is not accessible to readers, so clear information about how to prepare this should be included.

There is no information about the sources of the disinfectants used and there is no way to make check solubility or the recommended concentrations. Please add the sources and links to the specification sheets.

The MS will benefit for language revision. 

Author Response

Poland, May 1, 2019

Dear Reviewer,

We are very grateful for your time and valuable remarks, which will certainly help us to improve the quality of the manuscript.

The changes introduced in the text of the manuscript have been marked with yellow highlights.

Detailed response to the review is presented below:

Reviewer 2

The statistical analysis is not clear. As per earlier comments, please provide the model used for MANOVA stating the dependent and independent variables. Was there any repeat measurements in the design?

It has been corrected once again – lines 225-260

The Polish Standard PN-EN 1276:2010 is not accessible to readers, so clear information about how to prepare this should be included.

The recipe for the preparation of hard water has been given in lines 93-98

There is no information about the sources of the disinfectants used and there is no way to make check solubility or the recommended concentrations. Please add the sources and links to the specification sheets.

So far, in all our works of this type, the Reviewers and Editors have stated that commercial names of disinfectants and their producers should not be given in manuscript. For this reason, we have prepared tables 1 and 2 as it look like. However, to meet the Reviewer's request, we have included at the end of response to the review the table with full data about the tread names, manufacturers and the working concentrations recommended by the manufacturer (various units). Based on the data from this table, you can check all information about the tested disinfectants. However, we leave the decision to publish it in manuscript at the discretion of the Reviewer and Editor. Please, inform us what we should do.

The MS will benefit for language revision. 

We sent the manuscript for correction in a professional language office by a translator in the field of biological sciences, but he introduced only a few minor grammatical corrections.

Table 1. Characteristics of disinfectants

Group of   disinfectants

Trade name

Active substance

Manufacturer

Working solution   concentration

Quaternary ammonium compounds

Prontech

benzyl-C12-18-alkydimethylammonium chlorides

Rolvet

2 g/l

Quatosept

benzyl-C12-16 alkyldimethyl chlorides

Galvet

2,5 ml/l

Sansept   0200

didecyldimethylammonium chloride, benzyl-C12-16-alkyldimethyl   chlorides

Sanechem

3 ml/l

Oxidizing agents

Huwa San   TR-50

hydrogen peroxide, silver nitrate

DESIO

10 ml/l

Hysepta M-1

perlactic acid

Galvet

5 ml/l

Peroxat

peracetic acid, hydrogen peroxide

Agro-trade

5 ml/l

Virkon

bis (sulphate) bis (peroxymonosulfate) pentapotassium, benzenesulfonic acid, C10-13 alkyl   derivatives, sodium salts, malic acid, sulfamic acid

Naturan

20 g/l

Chlorine compounds

Armex 5

chlorine dioxide

Mexeo

10 ml solution 2000 ppm/l

Calcium hypochlorite

hypochlorous acid calcium salt

Chem Point

2 g/l

Iodine compounds

Jodat

iodine

Agro-trade

5 ml/l

Rapicid

iodine

Pfizer

10 ml/l

Nanoparticles

Nanocopper

nanocopper

Ecoworld

25 ppm

Nanosilver

nanosilver

Ecoworld

25 ppm

 Best regards,

Krzysztof Skowron